# Evaluation of the Physicochemical and Structural Properties and the Sensory Characteristics of Meat Analogues Prepared with Various Non-Animal Based Liquid Additives

**DOI:** 10.3390/foods9040461

**Published:** 2020-04-08

**Authors:** Gihyun Wi, Junhwan Bae, Honggyun Kim, Youngjae Cho, Mi-Jung Choi

**Affiliations:** 1Department of Food Science and Biotechnology of Animal Resources, Konkuk University, Seoul 05029, Korea; dnlrlgus@hanmail.net (G.W.); royal__style@naver.com (J.B.); moonjae@konkuk.ac.kr (Y.C.); 2Department of Food Science and Biotechnology, Sejong University, Seoul 05006, Korea; vollry@nate.com

**Keywords:** meat analogue, liquid additives, soy protein isolate, lecithin, emulsion

## Abstract

This study investigates the effects of various non-animal-based liquid additives on the physicochemical, structural, and sensory properties of meat analogue. Meat analogue was prepared by blending together textured vegetable protein (TVP), soy protein isolate (SPI), and other liquid additives. Physicochemical (rheological properties, cooking loss (CL), water holding capacity (WHC), texture and color), structural (visible appearance and microstructure), and sensory properties were evaluated. Higher free water content of meat analogue due to water treatment resulted in a decrease in viscoelasticity, the highest CL value, the lowest WHC and hardness value, and a porous structure. Reversely, meat analogue with oil treatment had an increase in viscoelasticity, the lowest CL value, the highest WHC and hardness value, and a dense structure due to hydrophobic interactions. SPI had a positive effect on the gel network formation of TVP matrix, but lecithin had a negative effect resulting in a decrease in viscoelasticity, WHC, hardness value and an increase in CL value and pore size at microstructure. The results of sensory evaluation revealed that juiciness was more affected by water than oil. Oil treatment showed high intensity for texture parameters. On the other hand, emulsion treatment showed high preference scores for texture parameters and overall acceptance.

## 1. Introduction

Increasing concerns about wellbeing in ethical, social, health, and ecological aspects have resulted in an increase in the number of vegetarians. The replacement of meat protein with plant protein has therefore become an important research topic [1,2]. However, meat analogues are still known to be different from meat in terms of mouthfeel, texture, taste, and flavor [3,4].

Many researchers have reported that fat plays a major role in meat or meat analogue quality such as juiciness, tenderness, mouthfeel, and flavor release of the product [5,6,7,8,9]. However, vegetable oils differ considerably in their physicochemical properties from animal fats, and this can therefore negatively affect juiciness and texture parameters [10,11].

Many studies have reported that an emulsion can be used as fat replacement in meat products owing to its positive effect on texture, water holding ability, and reduced fat content. Pietrasik et al. [12] reported that beef steak injected with 20% oil-in-water emulsion shows a decreased shear force and a high score of juiciness and tenderness in sensory evaluation. Serdaroğlu et al. [13] reported that using double emulsion as a beef fat replacement results in high water holding capacity (WHC), but decreased hardness. These results are due to not only the water content in the emulsion, but also depend on the type of oil and emulsifier [14].

An emulsion needs to amphiphilic emulsifier because it is a mixture of immiscible liquids such as oil and water. Emulsifiers have both a hydrophilic head group and lipophilic tail group, which can adsorb at the water–oil interface and act to lower interfacial tension. In addition, it can interact with the protein surface and cause structural changes in the protein, which affects the quality of meat protein [15,16]. Especially concerning soy protein isolate (SPI) and lecithin which are used as natural emulsifiers, Wang et al. [15] stated that heat-induced SPI affects the myofibrillar protein (MP), increasing the WHC, hardness, and springiness by forming a stable mixed gel by enhanced hydrophobic interaction and hydrogen bonds. On the contrary, Xia et al. [16] reported that the addition of lecithin to MP results in decreased hardness and springiness while the WHC increases as it increased hydrophobic interactions and hydrogen bonds, but decreased disulfide bonds. However, there are very few studies using emulsions and emulsifiers as a fat substitute for vegetable meat.

This study, therefore, aimed to analyze the effect of various non-animal-based liquid additives and their emulsifiers such as water, canola oil, O/W emulsion, SPI, and lecithin in meat analogues so as to compensate for the shortcomings of meat analogues.

## 2. Materials and Methods

### 2.1. Materials

Textured vegetable protein (TVP, SUPRO^®^ MAX 5050 and SUPRO^®^ MAX 5010, DuPont Korea, Seoul, Korea) and binder (GRINDSTED^®^ Meatline 2714, DuPont Korea, Seoul, Korea) were selected as the meat analogue base. Textured vegetable protein (TVP) contains 55%–60% of SPI, 40%–45% of wheat gluten and wheat starch. Furthermore, the binder is a mixture of egg white powder, glucose, soy protein, locust bean gum, carrageenan and guar gum (supplied information by DuPont Korea). SPI (90% of protein on a dry matter basis, Avention, Incheon, Korea) and lecithin (from soybean, Samchun, Seoul, Korea) were used as emulsifiers. Canola oil was purchased from Samchun (Seoul, Korea).

### 2.2. Sample Preparation

The flow diagram for manufacturing the meat analogue is described in Figure 1. The base of the TVP matrix was prepared using TVPs, SPI, and binder, following which the various non-animal-based liquid additives were added. A total of 100 g of TVP was immersed into water (10 times in volume) for hydration for two hours. Thereafter, the hydrated TVP was dehydrated in a centrifugal dehydrator (ws-6600, Hanil Electric, Seoul, Korea) for 5 min at 1200 rpm. Further, 100 g of swollen and dehydrated TVP was mixed with different quantities of liquid additives. The mixing ratio and ingredients are described in Table 1. Each of the liquid additives was individually prepared before blending with the processed TVP. SPI and lecithin (1% by weight) were dissolved in water and canola oil, respectively and stirred for 12 h. O/W emulsion and OW additives were both prepared by mixing together the same concentrations of water, oil, SPI, and lecithin. The OW additive was not pre-emulsified like the O/W emulsion preparation. The emulsion preparation required 1% (*w*/*w*) SPI to be dissolved in distilled water as the aqueous phase and 1% (*w*/*w*) lecithin to be dissolved in canola oil as the oil phase. The emulsification process entailed the two phases to be first mixed at 4:6 (oil phase: aqueous phase) of weight ratio and homogenized at 14,000 rpm for 5 min using a high-speed homogenizer (T25 digital ULTRA-TURRAX^®^, IKA, Staufen, Germany). This emulsion was then emulsified further using a high-intensity ultrasonicator with 75% power at 150 W capacity for 1 min (SONOPLUS HD2200, Bandelin^®^, Berlin, Germany).

All prepared liquids were mixed with 100 g of processed TVP, and then blended for 60 s using a hand blender (550 W, Multiquick 3 Vario, Braun, Kronberg im Taunus, Germany). The TVP matrix was molded in a cylindrical mold (30 mm of diameter and 25 mm of height). The matrix was then cooked in an oven (M4207, Simfer, Istanbul, Turkey) at 180 °C for 14 min and cooled to room temperature for 30 min before being used for further analysis.

### 2.3. Dynamic Rheological Properties

The rheological properties of the TVP matrix were measured with a rheometer, (MCR 302, Anton Paar, Graz, Austria) comprising of parallel-plate geometries (diameter 25 mm, gap 2.5 mm) and 2.8 g of the dough was molded in a cylindrical mold and trimmed with a spatula. The strain sweeps test was performed from 0.01% to 100% strain at 10 rad/s to obtain linear viscoelastic regions while an angular frequency sweep test was carried out from 1 to 100 rad/s at 0.1% strain at 25 °C. The storage modulus (G′), loss modulus (G″) and tan δ were recorded continuously to assess the rheological properties of the TVP matrix.

### 2.4. Cooking Loss (CL)

Cooking method and conditions were determined based on the study done by Pathare and Roskully [17]. The internal temperature of the meat analogue was measured by inserting thermocouples and memorizing the value in connected data loggers (Data Acquisition-MX 100, Yokogawa, Japan). The cooking conditions were set at a temperature of 180 °C for 14 min depending on the temperature of the meat analogue in its center reaching 80 °C. After cooking, the samples were cooled at room temperature for 30 min. CL was calculated as the percentage weight difference between the dough before cooking and after cooking, using the following formula:CL (°C) = (W_1_ – W_2_)/W_1_ × 100

W_1_: weight of meat analogue dough (g).

W_2_: weight of cooked meat analogue (g).

### 2.5. Water Holding Capacity

WHC was measured by modifying a method described previously [18]. Briefly, 2 g of the dough and cooked samples of the meat analogue were put in a 15 mL conical tube with gauze underneath. The tube was then centrifuged at 3000 rpm for 10 min at 35 °C (centrifuge-1736R, LaboGene, Daejeon, Korea). WHC was calculated by comparing the weight of the samples before and after centrifugation, using the following formula:WHC (%) = (W_2_/W_1_) × 100

W_1_: weight of meat analogue before centrifugation (g).

W_2_: weight of meat analogue after centrifugation (g).

### 2.6. Texture Profile Analysis (TPA)

TPA was measured by modifying a method described previously by Lin et al. [19] using a texture analyzer (CT3, Brookfield Engineering Labs Inc., Stoughton, MA, USA). The samples were shaped like a cylindrical column (30 mm in diameter and 25 mm in height). They were compressed to 50% deformation of its original thickness and were measured under the conditions of test speed (2.5 mm/s) and trigger load (100 g). A cylindrical probe (38.1 mm in diameter) was used for the test and the hardness, cohesiveness, springiness, and chewiness data were recorded.

### 2.7. Visible Appearance

The appearance of the meat analogue was assessed by adding 25 g of various liquid additives. The external and internal appearance was filmed with a digital camera (α350, Sony, Tokyo, Japan) and the characteristics were observed.

### 2.8. Color Analysis

The color of the meat analogue was analyzed by adding 25 g of the various liquid additives. The color of the inner cross-section from the cooked meat analogue was determined using a colorimeter (Chroma Meter CR-400, Konica Minolta, Tokyo, Japan). The color analysis results were expressed according to the Commission International de l’Eclairage (CIE) system and reported as Hunter L* (lightness), a* (redness), and b* (yellowness).

### 2.9. Microstructure

The microstructure of the meat analogue was obtained by a field emission scanning electron microscope (TM4000Plus, Hitachi, Seoul, Korea) with an accelerating voltage of 15 kV. The micrograph of the samples was taken in 100× and 300× magnification. Sample preparation was performed according to the method suggested by Samard and Ryu [4]. Briefly, the cooked samples were cut into thin slices (approximately 10 × 10 × 1 mm^3^) and were frozen at −100 °C for 24 h in a deep freezer (CLN, NIHON-FREEZER, Tokyo, Japan). Frozen samples were dried in a freeze-dryer (FDCF-12012, Operon, Gyeonggi-do, Korea) at a pressure of 5 Pa and a temperature of −80 °C for 48 h.

### 2.10. Sensory Evaluation

The meat analogue samples added with water, canola oil, O/W emulsion, and OW treatments were evaluated by ten experienced estimators who were graduate students and staff from the Department of Food Science and Biotechnology of Konkuk University. Sensory evaluation was conducted in individual booths on the basis of firmness, elasticity, stickiness, compactness, roughness, soy taste, oil taste, juiciness, and overall acceptance. A seven-point scoring test was used for evaluating the parameter intensity and sensory preference (7: very strong and good, 1: very weak and unacceptable). The Institutional Review Board (IRB) approved the consent procedure for sensory evaluation (nos. 700355-201901-HR-294). The written consents from all the participants were acquired before conducting the sensory evaluation.

### 2.11. Statistical Analysis

All experiments were analyzed using SPSS statistics (ver. 24.0, SPSS Inc., Chicago, IL, USA) except for sensory evaluation. The significance of the results was analyzed using one-way ANOVA and Duncan’s multiple range test which were conducted at the *p* < 0.05 level to verify the statistical significance of each sample.

## 3. Results and Discussion

### 3.1. Dynamic Rheological Properties

The measurement of the dynamic rheological properties is important to control the quality of processed meat product or meat analogues. Although there are many studies reporting on TVP protein properties regarding the extrusion method and modified vegetable proteins [2,4,19], there are few analyses considering the physical properties of emulsifier effects, or considering preformulated emulsions to improve their properties. Therefore, in order to investigate the effect of emulsifier or different liquid additives on TVP matrix, firstly, we tried to analyze the rheological properties of non-heated meat analogues using a Rheometer^®^.

As shown in Figure 2, the values of G′ and G″ were seen to decrease with increasing concentration of liquid additives in all the treatments. The value of tan δ tended to increase, which meant that the degree of decrease of the G′ value was larger than the one for the G″ value, except for oil treatment. Therefore, with an increasing concentration of liquid additives, the non-heated meat analogue acquires the viscosity properties (liquid-like) of TVP except for oil treatment. A review report by Song and Zheng [20] explained that water has an important role in controlling the viscoelastic properties of dough, and as water content increases, the G′ and G″ values decrease due to a lubricant effect. Likewise, Roccia et al. [21] reported that free water in the food system is known to reduce the elastic and viscous properties of dough due to its lubricating action. It is therefore supposed that the increase in liquid, especially free water, causes a lubricating action in the TVP dough.

The tan δ value of oil treatment without lecithin was seen to decrease with increasing oil concentration, although the G′ and G′′ values decreased with an increasing oil concentration similar to the other treatments. As the oil concentration was increased in the dough, the oil binding to the hydrophobic amino acid of proteins by hydrophobic interactions resulted in the formation of a uniform gel network [11], and it is supposed that the elastic behavior of dough increased significantly. Song and Zheng [20] also reported that an addition of oil promotes the aggregation of gluten and plays a role in increasing the elastic behavior of dough. Therefore, the values of G′ are relatively higher in oil treatment than in water treatment.

The values of G′, G″, and tan δ (G″/G′) of TVP were slightly increased after the addition of SPI in the water as a liquid additive. Studies done by Roccia et al. [21] have indicated that SPI can greatly uptake water owing to the hydrogen bond of SPI. Therefore, its high absorption of water contributes to its elastic properties, resulting in a solid-like substance [21]. Wang et al. [15] has reported that SPI, when mixed with wheat gluten, reduces the water availability caused by the disulfide bonds of wheat gluten. The rheological changes, therefore, describe the relative distribution of “elasticity” as compared to “viscosity” during the TVP gel matrix formation when mixed with water and SPI. This is presumed to be due to a reduction in lubricating action and water availability by having a higher content of bound water in the SPI.

The G′ and G″ values of oil-lecithin-added meat analogues were significantly decreased when compared to only oil-added samples resulting in an increased tan δ value (*p* < 0.05). The elasticity and viscosity properties of oil-lecithin-treated meat analogues become weak owing to the larger tan δ value. Lecithin is an amphiphilic surfactant, therefore, the hydrophilic region (choline and phosphate group) might be conjugated to water molecules or the hydrogen group in protein, while the hydrophobic region (monosaturated or saturated fatty acid) might be conjugated to oil molecules by hydrophobic interactions. When all the components are mixed, lecithin is adsorbed at the interface between water and oil, causing the mixture to become a more emulsified system, which is homogenously dispersed resulting in high lubrication properties [16], and it is the reason for an increase in the tan δ value. Azizi et al. [22] also reported that the extensibility of the flat bread dough decreased with the addition of lecithin.

The tan δ of emulsion or OW treatments were the highest among all treatments. This indicates that the addition of water and oil affect the rheological changes ranging from “elasticity” to “viscosity” markedly. However, there was no significant difference in the tan δ value between the emulsion-added TVP and the OW-added TVP.

### 3.2. Cooking Loss

CL presents the degree of meat shrinkage during cooking, which is an important indicator to evaluate the meat quality related to the juiciness and yield of the final product. In general, the CL of processed meat products is affected by preparation parameters like constituents of composite materials. Therefore, CL of meat analogues was determined depending on the various liquid additives, and has been shown in Figure 3.

The CL of water-added meat analogue with or without SPI was the highest among all treatments. As the amount of water increased, CL of meat analogues showed a typical increase from 12.5% to 14.5% regardless of SPI addition. The results are in accordance with those indicated by the researchers who cooked meat analogues at 70 °C for 150 min under sous-vide conditions [23]. However, there was no influence of SPI on CL of meat analogues, except for a loss of 15 g content. Although elasticity properties of meat analogues were enhanced when compared to the previous results by water uptake of SPI, SPI contributed to no significant improvement in CL. Wang et al. [15] reported that when SPI was heated at a high temperature, the hydrophobic amino acid residues get exposed by reducing the β-sheet content. Therefore, the heated-SPI cannot play a role in water uptake resulting in high CL.

On the other hand, CL of oil-added meat analogues (without lecithin) was seen to decrease with increasing oil concentration until the lowest value was attained. Interestingly, the lecithin influence on the CL of meat analogue was reflected by the significant increase when compared to oil treatment without lecithin (*p* < 0.05). Without lecithin, the oil induced hydrophobic interactions between protein and oil, or between protein and protein, enhancing the elastic behavior and gel strength. This means that the retained water in meat analogue mixture could be entrapped into gel structures bounded by hydrophobic interactions. This may explain why the CL of oil containing meat analogue was the lowest among all the samples. Lecithin is amphiphilic and attracts both water and fatty substances. Therefore, when lecithin was mixed with the oil and base of TVP, the hydrophobic region in lecithin may have bound to the canola oil or protein by hydrophobic interactions [16,24,25]. Meanwhile, the choline or phosphate groups of the hydrophilic part of lecithin binds to retained water by hydrogen bonds in the hydrated TVP. Huang et al. [26] reported that hydrogen bonds had a significantly negative correlation with gel strength. This study showed that hydrogen bonds were weakened as the temperature increased. Therefore, it is a possibility that once the water-oil-TVP protein network was set by lecithin, a high temperature caused an increase in hydrophobic interactions as decreasing hydrogen bonds caused an intense “shrinkage” of the network, resulting in a leakage of water.

The value of CL in emulsion-added samples presented was between the WS-added sample and the OL-added sample (*p* < 0.05). The CL of emulsion treatment was slightly higher than that in the OL treatment. Moreover, there was no difference in the CL values between the emulsion-added meat analogues and the OW-added meat analogues.

### 3.3. Water Holding Capacity

WHC is an important factor as it affects the quality and yield of fresh meat or its products [15]. In meat analogues, WHC represents the ability of protein to hold water and to form the protein gel network. The higher the WHC in meat analogues, the more enhanced the juiciness. Therefore, WHC of meat analogues was investigated depending on the kind of liquid additives before heating and after heating.

Figure 4 shows the effect of liquid additives on the WHC of non-heated and heated-meat analogues. Overall, the WHC of non-heated meat analogues was seen to be lower than that of the heated-meat analogues for all treatments. It was supposed that the heating process causes an enhanced gel network formation of TVP by hydrophobic interactions, which could retain more water when compared to the non-heated TVP system. There are numerous studies that have reported that the heating process can improve the gel network of a soy-based protein [15,23,26]. These studies have justified that once the protein structure of TVP has unfolded and the hydrophobic groups have been exposed by heating, the hydrophobic interaction between proteins is induced. Accordingly, the WHC value of cooked meat analogue is found to be higher than that in the dough condition.

The WHC of the water-added meat analogues, was seen to be affected by SPI under non-heated conditions (*p* < 0.05). Roccia et al. [21] explained the water syneresis of vegetable proteins between SPI and gluten proteins. They reported that the increase of soy addition dramatically decreased free water for syneresis in a mixture of gluten protein and soy protein resulting in high water holding capacity.

The WHC of the oil-added meat analogues was higher than that of the other meat analogues both under non-heated and heated conditions. The high WHC of the former may be explained by an interaction between the hydrocarbon side chains of oil and the hydrophobic amino acids of TVPs. This hydrophobic interaction causes gel matrix formation, which has a greater ability to entrap water. Therefore, the oil addition affects the WHC of the TVP matrix. On the other hand, WHC of lecithin was significantly decreased depending on the concentration of the oil content (*p* < 0.05). When lecithin was added into this system, the hydrophilic head would have bound to water molecules by hydrogen bonds, while the hydrophobic phospholipid fatty acid chains could be bound to the hydrophobic part of TVP or oil. Here, the holding ability of water was weakened with an increasing concentration of oil and lecithin [16]. This could be a result of lecithin blocking the crosslinks between oil and protein. Instead, lecithin strongly attracts the contained water molecules with hydrogen bonds, resulting in water exposure [27]. Therefore, this water may be released after centrifugation for measurement of WHC. This result is consistent with the observed CL. Since hydrogen binding between lecithin and water molecules are destroyed by an increasing temperature [15], WHC of lecithin-oil-TVP was decreased by an increasing concentration of lecithin and oil after heating.

The WHC of emulsion and OW-treated meat analogues was enhanced by the heating process. The WHC values for non-heated meat analogues, however, showed no difference between emulsion and OW treatments. However, after heating, the WHC of OW-treated meat analogues was seen to be slightly higher than that of emulsion-treated meat analogues.

### 3.4. Texture Profile Analysis

Texture of meat analogues is an important factor to mimic the organoleptic taste of muscle. Figure 5 shows the textural parameters (hardness, cohesiveness, springiness, and chewiness) of meat analogues on treatment with different liquid additives. All TPA parameters were the lowest on water treatment, whereas the highest was on oil treatment (*p* < 0.05). This result is consistent with the rheological properties. Lin et al. [19] reported the hardness, cohesiveness, chewiness, and gumminess decreased as the moisture content increased. This means that the higher water contents formed more softened meat analogues which is the same as our results. On the other hand, Barbut and Marangoni. [11] studied that the smaller size of canola oil can connect protein–protein interaction resulting in a firmer meat product owing to its larger surface area among the protein matrix. These reports were consistent with our results where canola oil formed a compact protein gel network of TVP matrix since the oil globule binds to the hydrophobic amino acids in the protein by a hydrophobic interaction [11].

With increasing concentrations of liquid additives, hardness, cohesiveness, springiness, and chewiness decreased in all treatments (*p* < 0.05). This result was inevitable due to the reduction of TVP content in the same weight of meat analogue as the concentration of liquid additive increased [19,28]. All TPA parameters were seen to be significantly increased in meat analogues with water and SPI addition, while there was a significant decrease in meat analogues with oil and lecithin addition (*p* < 0.05). These results indicate that SPI might improve the gel properties of the TVP complex. On the contrary, lecithin may inhibit the gel matrix formation by hydrophobic interactions, thereby enhancing their lubricant effect by interacting between the water molecules and the polar ionic head groups. Wang et al. [15] reported that SPI enhanced the hardness of meat protein due to an increase in hydrophobic interactions. Regarding the role of lecithin, Xia et al. [16] investigated how it inhibits the disulfide binding of the cross-linking of proteins, resulting in a decrease in the hardness of meat protein. Therefore, it is considered that SPI and lecithin may have a similar effect on the TVP protein matrix.

Interestingly, all TPA parameters of the matrix formed by the addition of both oil and water with SPI and lecithin showed intermediate values between WS treatment and OL treatment, but were closer to WS treatment. Accordingly, the TPA parameters seem to be more influenced by the continuous phase (WS) than the dispersed phase (OL) of the added liquid additives. All TPA parameters of the emulsion and OW treatments showed intermediate values between WS and OL treatment because they included a mixture of the continuous phase (WS) and the dispersed phase (OL). However, there was no significant difference between the emulsion and OW treatments.

### 3.5. Color Analysis

The color measurement according to the type and concentration of liquid additives is shown in Table 2. As the content of liquid additives increased, the lightness (*L**) values of all treatment was seen to significantly increase, except for OL treatment. The small globules such as water or oil cause more light reflection, and they increase the *L** values [11,29]. Youssef and Barbut [30] reported that the *L** values decreased with an increasing protein content in meat products, whereas other researches have reported [29] that *L** values increased with rising canola oil content. Our results showed similar trends where the *L** value was increased during the decrease of TVP contents and an increase in oil content [30]. Especially, the *L** values of emulsion treatment was the highest among all the treatments because of its milky color. Lee et al. [31] reported that the pork patty added with nanoemulsion has the highest *L** values due to its white color. The *L** value of water-SPI-added samples was slightly decreased when compared to only water-added samples due to the increase in protein content [30], also *L** value of oil-lecithin-added samples was greatly decreased when compared to oil treatment due to the yellow-brownish color of the lecithin [16,31] (*p* < 0.05). However, there was no effect from the emulsifiers on the color of the TVP matrix in the emulsion treatment. The redness (*a**) value of all treatments was overall lower compared to the meat products due to a low concentration of myoglobin pigment [4]. As the content of liquid additives increased, the *a** value of all treatments was seen to significantly decrease, except for water treatment. Especially, the *a** value of emulsion treatment showed the lowest value (*p* < 0.05) owing to the milky color of the emulsion [31].

### 3.6. Visible Appearance

The external and internal appearance of meat analogues after cooking has been presented in Figure 6. The external appearance showed no difference on observation with water addition and water with SPI treatment. The water-added samples with or without SPI were found to have a rough surface when compared to the other samples due to the high evaporation of moisture on the surface. However, the internal appearance of the water-added samples with SPI obviously appeared more homogeneous and finely structured when compared to the only water-added samples. This therefore proved that the SPI addition might absorb the water molecules to form the fine gel matrix [15]. Furthermore, after cooking, the volume of the meat analogue added with water was slightly reduced due to evaporation of the water [4].

Oil-added samples appeared bright yellow in color with a fine surface texture and volume expansion when compared to lecithin-oil-added samples, which showed cracks. The interior of lecithin-oil-added samples showed void pores when compared to the only oil-added sample. This led to the supposition that the fine gel matrix in oil treatment may hinder the water evaporation and therefore, the captured water molecules during cooking may cause the volume expansion of the meat analogues [4].

The exterior of the meat analogues with emulsion and OW treatment did not show a rough surface or cracks, unlike the ones with WS treatment and OL treatment. On the other hand, the internal appearance of the emulsion and OW-treated meat analogues showed a porous and rough structure like those of the ones treated with water.

### 3.7. Microstructure

For the structural observation, the amount of liquid additives was fixed at 25 g for all treatments. Microstructure of the six varied TVP matrices after cooking were observed under a scanning electron microscope with a magnification of 100 times (Figure 7) and 300 times (Figure 8). The water-added samples without SPI showed inhomogeneous pore distribution and a large air cell partly (see white arrow in Figure 7A). Meanwhile, SPI-water-added samples displayed more homogeneous and very distinct fibrous pores in the matrix, which exhibited a more sponge-like structure as compared to the others (Figure 8). Wang et al. [15] found a more ordered network between meat protein and SPI by obvious cross-linked strands, which can hold more water to form a fine gel matrix. This report was consistent with our results in meat analogues, and verified that the SPI could absorb water upon hydration and entrap the water molecules inside the gel matrix to form a finer structure of the gel [15], which probably contributes to higher WHC (Figure 4) and hardness (Figure 5) and more elastic properties (Figure 2).

The oil-added samples without lecithin formed a continuous network comprising of a very fine and tightly connected matrix by hydrophobic interaction between TVPs and oil. On the contrary, the microstructure of the lecithin-oil-added samples showed structural similarities to the water-added samples, having a large air cell (see red arrow in Figure 7D) and was partly cracked, although there was no additional water as a liquid additive. Xia et al. [16] mentioned that the hydrophilic head of lecithin can be bound to water molecules, which indicate that lecithin might attract the water, which was absorbed during the swelling process of TVP preparation before the addition of liquid additives. Thus, the hardness and WHC were decreased, and the cooking loss was increased with an increasing concentration of oil with the lecithin additive [16,32].

The microstructure of the emulsion-added TVP matrix was the most fine and the smallest pore size in the samples. At 100 times magnification, the emulsion-added sample shows an elongated shape of the pore—less than one of the water-SPI-added sample. When observed at 300 times magnification, the emulsion-added samples showed uniform round-shaped pores in the fibrous structure. The emulsions added to the TVPs may improve the homogeneous structure of the composites owing to pre-emulsification of liquid additives. Zhuang et al. [33] reported that the morphology of cooked meat batters applied with pre-emulsified sesame oil showed a homogeneous and compact structure and pores of semi-round shapes. Likewise, Jimenez-Colmenero et al. [34] reported that the microstructure of the frankfurters replaced with emulsion showed small cavities. It is presumed that the emulsifier in an O/W emulsion assists greater dispersion of the oil phase, which was well embedded in the TVP matrix. Comparatively, the OW-addition, which is a non-emulsified liquid additive, presented a coarser and larger air cell than the emulsion-added sample at 100 times magnification. This was also identified in the image obtained at 300 times magnification. Although the microstructures were different between the emulsion-added TVP and the OW-added one, there was no significant difference in the WHC, CL, and TPA parameters. Subsequent studies are needed to determine the difference between emulsion and OW treatment.

### 3.8. Sensory Evaluation

The sensory evaluation was carried out to analyze the differences in the meat analogues. As shown in Figure 9, intensity and preference score of juiciness were the highest in water treatment and the lowest in oil treatment. According to CL and WHC results, contrary to the expectation that the oil treatment with the lowest CL value and the highest WHC value had the highest juiciness, the juiciness of water treatment was found to be the highest. These results mean that juiciness is more affected by water than oil. Selani et al. [35] reported that a beef burger added with canola oil as a fat replacement enhanced the cohesiveness and springiness in its sensory attributes. According to a report by Kim et al. [25], in a sensory test of vegetable meat, MCT oil showed a positive correlation with compactness and springiness and also showed a negative correlation with juiciness and tenderness. On the other hand, water showed a positive correlation with juiciness and tenderness. In our research, water treatment had the lowest value in all TPA parameters (Figure 5) because of the inhomogeneous structure of the TVP matrix (Figure 7A). However, our results in the sensory test showed that water-treated samples had a high intensity of firmness, elasticity, and compactness in contrast to the TPA results and other research [25]. This reason is supposed to be that the high-density region of the TVP matrix (see yellow arrow in Figure 7A) resulted in a feeling of firmness to chew in terms of mastication.

The preference score of the oil taste at oil treatment was the lowest owing to its excessively oily feeling. The intensity of soy taste did not show any tendency based on oil and water content, and also did not show any tendency between intensity and preference.

Emulsion treatment showed low intensity scores in relation to texture parameters such as firmness, elasticity, stickiness, compactness, and roughness. Therefore, emulsion treatment had a high preference score in relation to texture and overall acceptance. Lee et al. [31] reported that the pork patty added with nanoemulsion showed the highest intensity score for juiciness and tenderness, while it showed the lowest intensity score for compactness and springiness in the sensory test. Kim et al. [25] also reported that meat analogue added with emulsion showed a high juiciness and tenderness, which caused high overall acceptability. Likewise, Pietreasik et al. [6] have proved that most consumers prefer tenderness to be a significant criteria for determining the overall acceptance and quality of meat. These studies are similar results to these results because the emulsion treatment showed the finest structure and the smallest pore size, and a homogeneous structure of the TVP matrix, owing to the emulsification process (Figure 8E), resulted in good texture.

In the sensory evaluation, a difference was seen between the emulsion-added sample and the OW-added sample owing to the differences in pore size (Figure 8E,F). This indicates that the OW-added TVP matrix was less homogeneous than the emulsion-added TVP matrix, resulting in a high intensity of firmness, compactness, and roughness.

## 4. Conclusions

The present study evaluated the physicochemical, structural, and sensory properties of meat analogues according to the type and concentration of various non-animal-based liquid additives. According to an analysis of CL, WHC, and TPA, SPI had a positive effect on the gel network formation of TVP matrix, whereas lecithin had a negative effect. The values of emulsion treatment are between water treatment added with SPI and oil treatment added with lecithin. There was no significant difference between the pre-emulsified mixture and the separated liquid additives in terms of physicochemical properties, but slight differences were seen in the structural properties and sensory characteristics. Especially, the sensory evaluation showed that juiciness was positively correlated with water, and the preference of texture parameters and overall acceptance were positively correlated with emulsion treatment.

The results of our study have thus effectively analyzed and presented the possibility of using certain liquid additives to improve the acceptability of meat analogues. The substitution of animal fat in meat analogues while maintaining its sensory and physicochemical profile to ensure its acceptability by vegetarians is a present day challenge for the food science industry. This work, if enhanced further, promises to successfully provide answers to this challenge.

## Figures and Tables

**Figure 1 foods-09-00461-f001:**
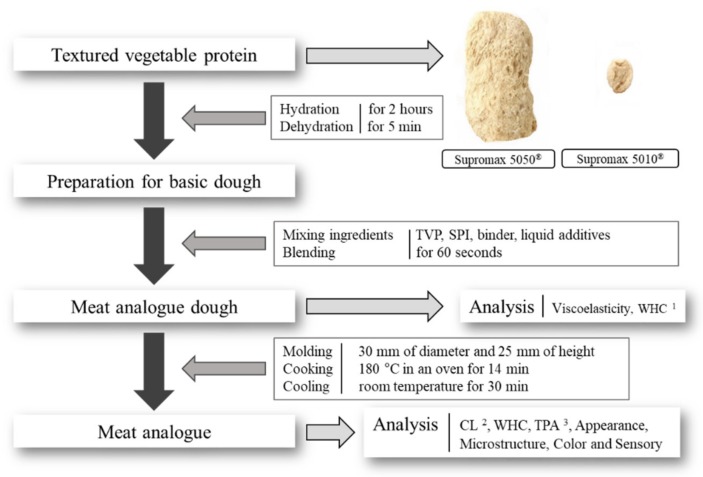
Flow diagram for manufacturing the meat analogue. ^1^ WHC: water holding capacity. ^2^ CL: cooking loss. ^3^ TPA: texture profile analysis.

**Figure 2 foods-09-00461-f002:**
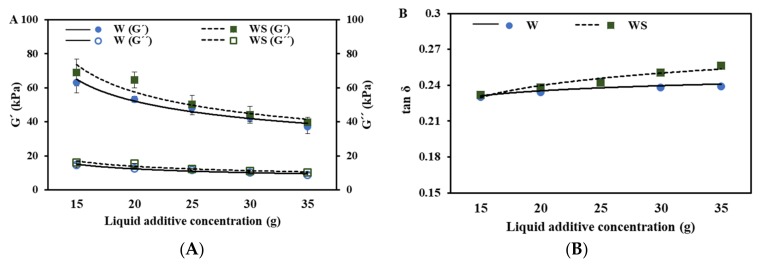
The differences in the rheological behavior (G’: the storage modulus; G´´: the loss modulus (**A**, **C** and **E**); tan δ: loss tangent (**B**, **D** and **F**)) of the meat analogue dough based on the liquid additives and emulsifier type.

**Figure 3 foods-09-00461-f003:**
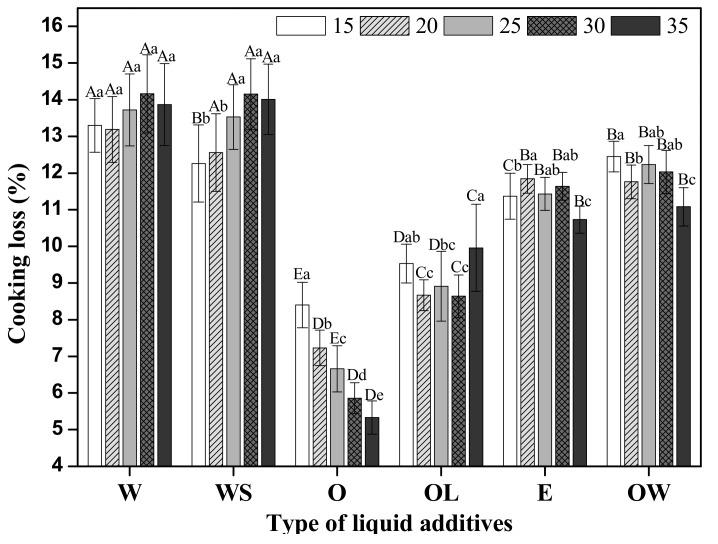
The CL value of meat analogues based on the type and concentration of liquid additives. ^a–e^ indicates significant differences in treatment concentrations (*p* < 0.05); ^A–E^ indicates significant differences in treatment types (*p* < 0.05).

**Figure 4 foods-09-00461-f004:**
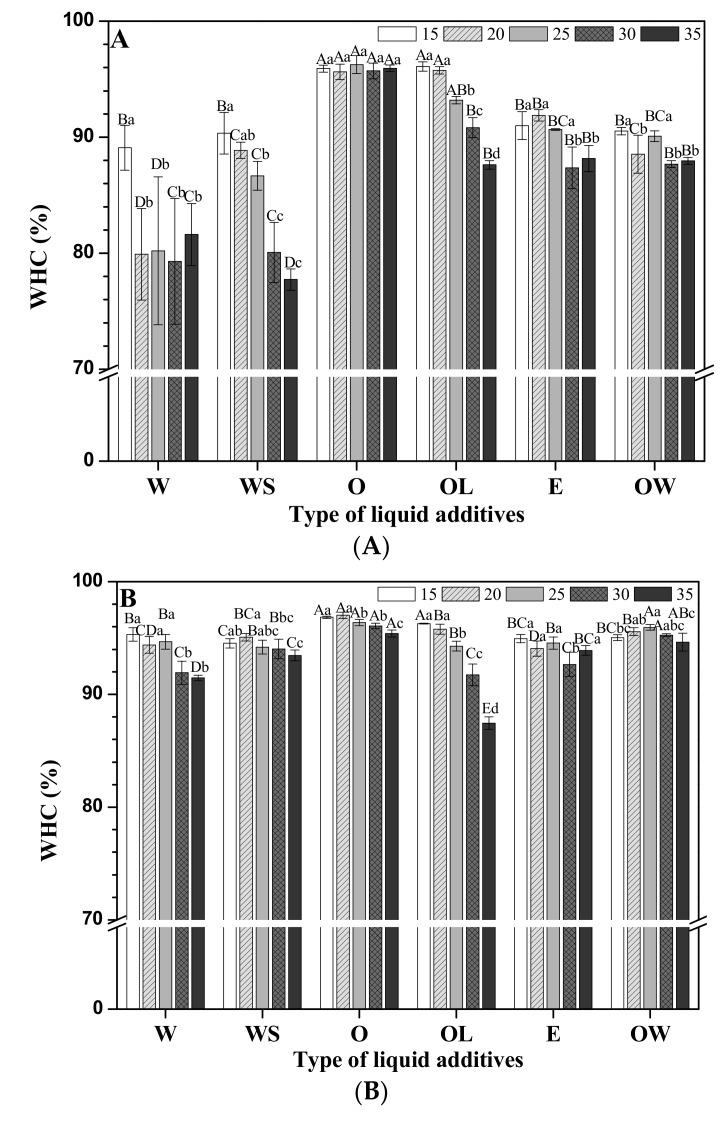
The WHC value of meat analogues based on the type and concentration of liquid additives in dough (**A**) and cooked (**B**) conditions. ^a–d^ indicates significant differences in the treatment concentrations (*p* < 0.05); ^A–E^ indicates significant differences in the treatment types (*p* < 0.05).

**Figure 5 foods-09-00461-f005:**
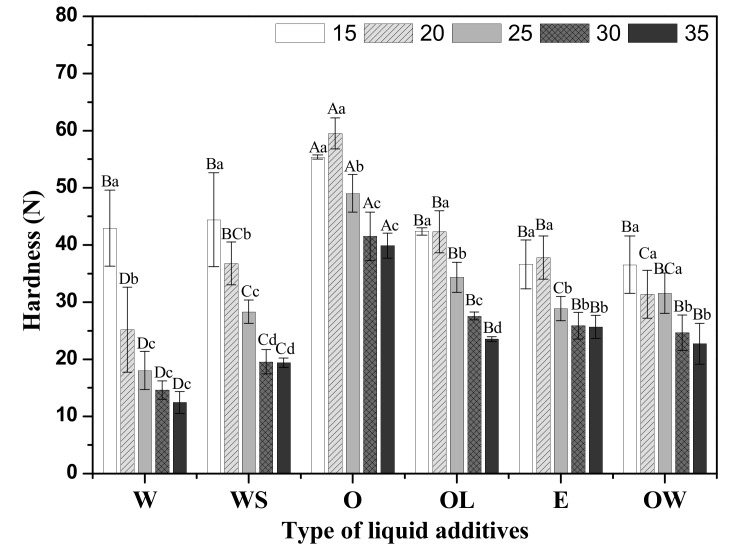
The TPA parameters of meat analogue based on the type and concentration of liquid additives. ^a–d^ indicates significantly different treatment concentrations (*p* < 0.05); ^A–^^E^ indicates significantly different treatment types (*p* < 0.05).

**Figure 6 foods-09-00461-f006:**
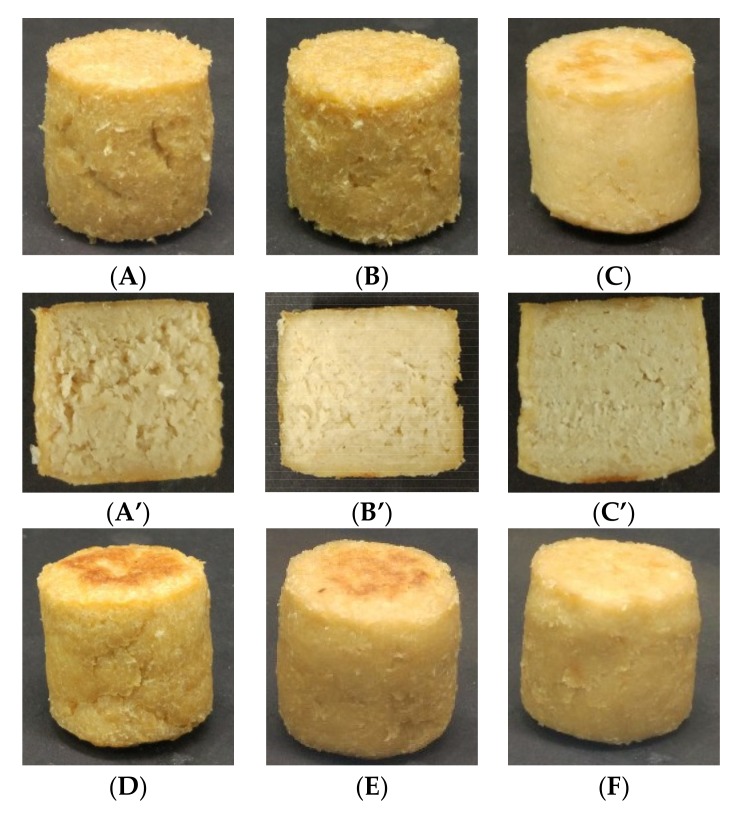
The external (**A**–**F**) and internal (**A´**–**F´**) appearance of meat analogue in six different liquid additives. (**A**) and (**A´**): water; (**B**) and (**B´**): water + SPI; (**C**) and (**C´**): canola oil; (**D**) and (**D´**): canola oil + lecithin; (**E**) and (**E´**): O/W emulsion; (**F**) and (**F´**): water + canola oil + SPI + lecithin.

**Figure 7 foods-09-00461-f007:**
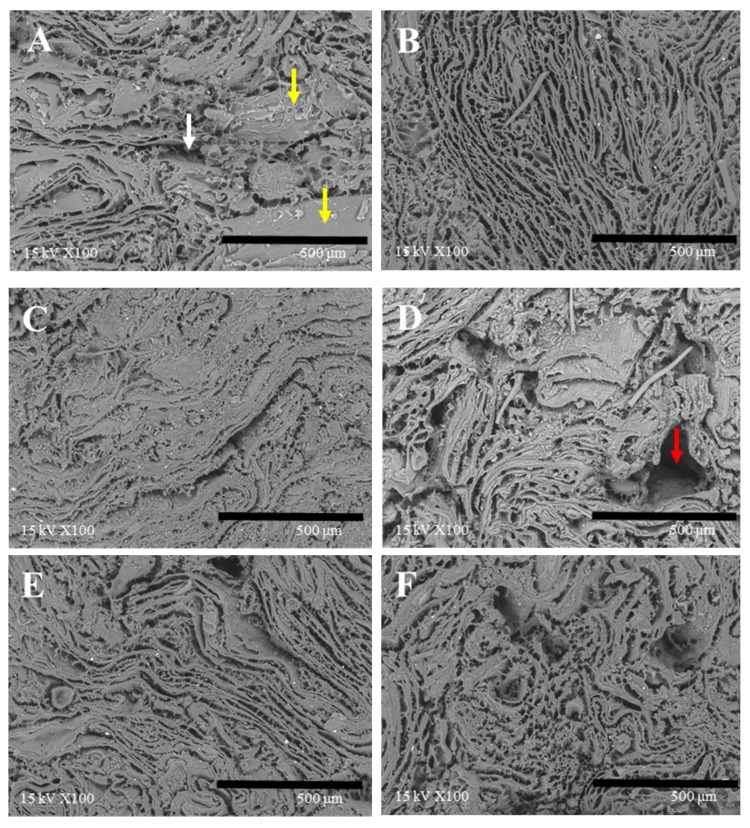
Scanning electron microscope images of samples (magnification 100 times). (**A**): water; (**B**): water + SPI; (**C**): canola oil; (**D**): canola oil + lecithin; (**E**): O/W emulsion; (**F**): water + canola oil + SPI + lecithin.

**Figure 8 foods-09-00461-f008:**
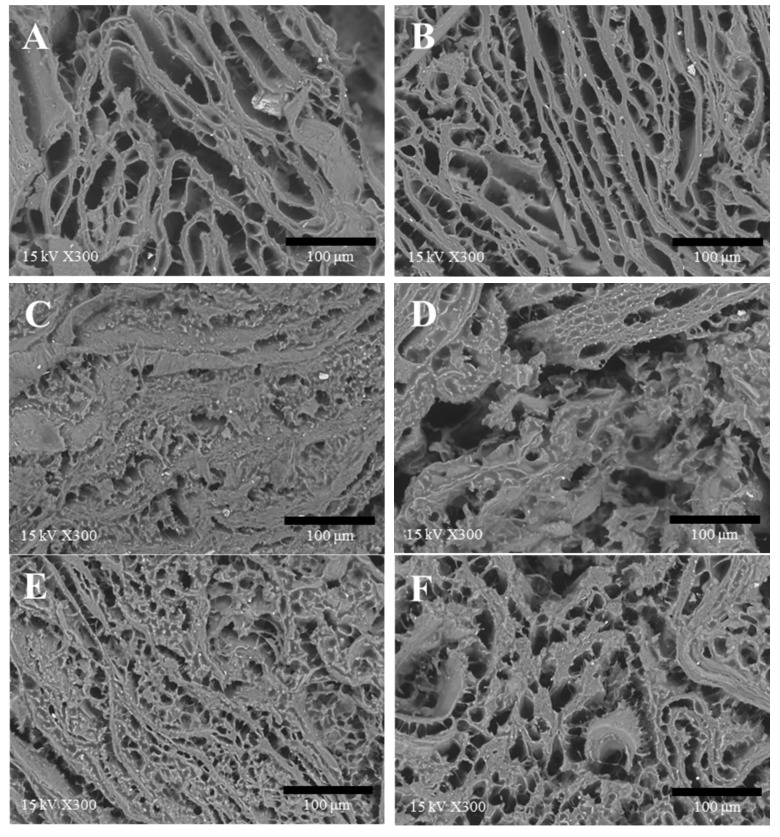
Scanning electron microscope images of samples (magnification 300 times). (**A**): water; (**B**): water + SPI; (**C**): canola oil; (**D**): canola oil + lecithin; (**E**): O/W emulsion; (**F**): water + canola oil + SPI + lecithin.

**Figure 9 foods-09-00461-f009:**
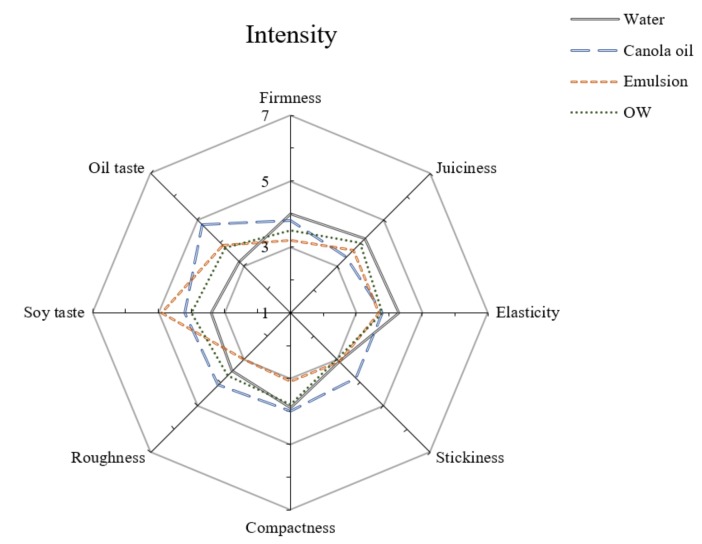
Sensorial profile of meat analogue based on various liquid additives.

**Table 1 foods-09-00461-t001:** Mixing ratios of additives in meat analogues.

Types of Liquid Additives ^3^	Mixing Ratios			
	Liquid Additive Concentration (g)	TVP ^1^ (g)	SPI ^2^(g)	Binder (g)
W	15	100	4.5	3.0
	20	100	4.5	3.0
	25	100	4.5	3.0
	30	100	4.5	3.0
	35	100	4.5	3.0
WS ^4^	15	100	4.5	3.0
	20	100	4.5	3.0
	25	100	4.5	3.0
	30	100	4.5	3.0
	35	100	4.5	3.0
O	15	100	4.5	3.0
	20	100	4.5	3.0
	25	100	4.5	3.0
	30	100	4.5	3.0
	35	100	4.5	3.0
OL ^4^	15	100	4.5	3.0
	20	100	4.5	3.0
	25	100	4.5	3.0
	30	100	4.5	3.0
	35	100	4.5	3.0
E ^5^	15	100	4.5	3.0
	20	100	4.5	3.0
	25	100	4.5	3.0
	30	100	4.5	3.0
	35	100	4.5	3.0
OW ^6^	15	100	4.5	3.0
	20	100	4.5	3.0
	25	100	4.5	3.0
	30	100	4.5	3.0
	35	100	4.5	3.0

^1^ TVP: textured vegetable protein. ^2^ SPI: soy protein isolate. ^3^ W: water, WS: water + SPI, O: canola oil, OL: canola oil + lecithin, E: O/W emulsion, OW: water + canola oil + SPI + lecithin. ^4^ One weight percent of emulsifier per added water or oil.^5^ Mixing ratio of pre-emulsified liquid additives (water with 1% SPI and oil with 1% lecithin is 6:4). ^6^ Same amount and ingredients as emulsion, but each of the compounds was not pre-emulsified.

**Table 2 foods-09-00461-t002:** Effects of the type and concentration of liquid additives on color of meat analogue.

Color				
Treatments	Concentration (g)	*L**	*a**	*b**
W	15	62.32 ± 4.02 ^Cc^	2.21 ± 0.23 ^ABb^	16.19 ± 0.62 ^Ac^
	20	62.61 ± 1.12 ^Dc^	1.91 ± 0.07 ^Bc^	16.25 ± 0.15 ^Bc^
	25	64.55 ± 2.47 ^Dbc^	1.93 ± 0.16 ^Ac^	16.47 ± 0.68 ^Ac^
	30	65.72 ± 1.71 ^Cab^	2.62 ± 0.25 ^Aa^	17.60 ± 0.39 ^Ab^
	35	67.81 ± 1.04 ^Ca^	2.62 ± 0.14 ^Aa^	18.22 ± 0.27 ^Aa^
WS	15	61.64 ± 0.90 ^Cb^	2.34 ± 0.03 ^Aa^	16.19 ± 0.26 ^Ab^
	20	61.58 ± 0.59 ^Db^	2.23 ± 0.06 ^Aab^	16.07 ± 0.17 ^BCb^
	25	62.41 ± 1.11 ^Eb^	2.10 ± 0.19 ^Abc^	16.25 ± 0.20 ^ABb^
	30	64.54 ± 0.93 ^Ca^	2.21 ± 0.23 ^Bab^	16.55 ± 0.77 ^Bab^
	35	65.38 ± 1.34 ^Da^	1.98 ± 0.15 ^Bc^	17.05 ± 0.76 ^Ba^
O	15	65.26 ± 0.52 ^Bd^	2.07 ± 0.14 ^BCa^	15.63 ± 0.30 ^Ba^
	20	67.64 ± 0.73 ^Bc^	1.90 ± 0.09 ^Ba^	15.84 ± 0.34 ^Ca^
	25	68.94 ± 1.34 ^Bb^	1.65 ± 0.17 ^Bb^	15.37 ± 0.44 ^Ca^
	30	72.12 ± 1.46 ^Aa^	1.28 ± 0.04 ^Ec^	15.69 ± 0.96 ^Ba^
	35	69.41 ± 0.73 ^Bb^	1.52 ± 0.23 ^Cb^	16.03 ± 0.52 ^Ca^
OL	15	65.96 ± 1.40 ^Ba^	2.26 ± 0.31 ^ABa^	16.02 ± 0.17 ^ABbc^
	20	65.87 ± 1.10 ^Ca^	2.23 ± 0.14 ^Aa^	16.14 ± 0.20 ^Bb^
	25	66.80 ± 0.19 ^Ca^	1.95 ± 0.17 ^Ab^	15.87 ± 0.29 ^Bc^
	30	65.77 ± 1.23 ^Ca^	1.93 ± 0.13 ^Cb^	16.20 ± 0.03 ^Bb^
	35	63.89 ± 0.66 ^Eb^	2.01 ± 0.26 ^Bab^	16.57 ± 0.07 ^BCa^
E	15	68.82 ± 0.62 ^Ac^	1.73 ± 0.04 ^Da^	16.36 ± 0.20 ^Aa^
	20	70.09 ± 0.83 ^Ab^	1.75 ± 0.08 ^Ca^	16.52 ± 0.23 ^Aa^
	25	70.75 ± 0.58 ^Ab^	1.48 ± 0.08 ^Bb^	16.31 ± 0.04 ^ABa^
	30	70.08 ± 0.41 ^Bb^	1.54 ± 0.10 ^Db^	16.36 ± 0.38 ^Ba^
	35	71.70 ± 0.54 ^Aa^	1.32 ± 0.02 ^Cc^	16.32 ± 0.28 ^Ca^
OW	15	68.52 ± 0.56 ^Ab^	1.98 ± 0.04 ^Ca^	16.19 ± 0.03 ^Ab^
	20	69.17 ± 0.97 ^Ab^	1.89 ± 0.05 ^Ba^	16.17 ± 0.15 ^Bb^
	25	70.51 ± 0.80 ^ABa^	1.65 ± 0.01 ^Bb^	16.17 ± 0.09 ^ABb^
	30	69.21 ± 0.73 ^Bb^	1.74 ± 0.05 ^Cb^	16.52 ± 0.09 ^Ba^
	35	70.51 ± 0.36 ^Ba^	1.51 ± 0.14 ^Cc^	16.13 ± 0.07 ^Cb^

^a–d^ indicates significant differences in treatment concentrations (*p* < 0.05). ^A–E^ indicates significant differences in treatment types (*p* < 0.05).

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
