# Peer review of "Evaluation of the Physicochemical and Structural Properties and the Sensory Characteristics of Meat Analogues Prepared with Various Non-Animal Based Liquid Additives"

_foods, 2020, doi:10.3390/foods9040461_

Round 1

Reviewer 1 Report

The ever-increasing fashion of not eating meat creates great new opportunities for the food industry to produce vegetarian food imitating meat products. This study investigates the effects of various non-animal based liquid additives on the physicochemical, structural, and sensory properties of meat analogues. The combination of the texture characteristics of the products with the study of taste, smell, texture would give a complete picture and value of the proposed food. The presented texture results are very valuable in the assessment of product characteristics, but do not reflect the full characteristics and value of the product. There is a lack of session results made by the panel evaluating the taste, smell and other characteristics of the product.

Comments to the article:

The scores of the Results and Discussion:

 In subsection 3. 4. “Texture profile analysis”:  Authors did not discuss the results of their research concerning the texture profile analysis in much detail. Only the results of one study were cited, which is not satisfactory at all. In subsection 3. 5. “Colour analysis” results of research were not discussed at all. These subchapters definitely need to be expanded and improved. Similarly in subsection 3. 6. “visible appearance” no discussion of the  results have not been made at all in relation to other available studies. Subchapter 3. 7. “Microstructure” needs to be developed, especially the part of discussion, unfortunately any single work obtained by other authors has been cited. Subchapter 3. 8 “Sensory evaluation” definitely  requires discussion and comparison with other research results, similarly as in the previous subchapters, not a single study was cited.  

The article definitely needs to be improved, especially the chapter “Results and Discussion”, where the authors did not refer to the results of other authors research. If the authors believe that the work is pioneering and unfortunately it is not possible to compare the results with other tests because no one did them, it should be emphasized at the beginning of the chapter "results and discussion"

Reviewer 2 Report

Dear Authors,
Please, introduce a procedure of sample preparation as a diagram(s)

In Reviewer opinion presented draft is too long: to much factors or to much experimental techniques. There are presented only results (for instance viscoelastic properties or texture) without any modeling or deeper discussion based on literature.

Round 2

Reviewer 1 Report

The new version of the article has been corrected taking into account the comments contained in the review of the first version of the article. In my opinion, the article is suitable for publication in the Foods Journal in this form.